# Green Space Exposure Association with Type 2 Diabetes Mellitus, Physical Activity, and Obesity: A Systematic Review

**DOI:** 10.3390/ijerph18010097

**Published:** 2020-12-25

**Authors:** Felipe De la Fuente, María Angélica Saldías, Camila Cubillos, Gabriela Mery, Daniela Carvajal, Martín Bowen, María Paz Bertoglia

**Affiliations:** 1Nursing Department, University of Chile, Santiago 8320000, Chile; felipedelafuente@med.uchile.cl (F.D.l.F.); masaldias@uchile.cl (M.A.S.); camifcug@gmail.com (C.C.); gabrielamerym@gmail.com (G.M.); dcarvajalpizarro@gmail.com (D.C.); 2Medical School, University of Chile, Santiago 8320000, Chile; mbowenv@gmail.com; 3School of Public Health, University of Chile, Santiago 8320000, Chile

**Keywords:** green spaces, diabetes mellitus, physical activity, obesity, overweight, systematic review

## Abstract

Type 2 diabetes mellitus (T2DM) is a public health challenge that must be addressed considering the large number of risk factors involved in its appearance. Some environmental risk factors are currently described as predictors of diabetes, with access to green spaces being an element to consider in urban settings. This review aims to study the association between exposure to green spaces and outcomes such as diabetes, obesity, and physical activity in the general population. A systematic review was carried out using the PubMed, Embase, and LILACS databases and other sources. The search strategy was carried out from October 2019 to October 2020. Cross-sectional and cohort studies were included. The article selection was made by a pair of reviewers, and data extraction was carried out using a data extraction sheet. The quality assessment of the included studies was carried out using a validated tool. Finally, 19 scientific articles were included in this review. Evidence supports that people and communities exposed to green spaces, especially in their neighborhood, reduce the risk of T2DM and reduce the risk of being obese and increase the likelihood of physical activity. The onset of T2DM can be moderated by using green spaces, improving physical activity levels, and reducing the risk of being overweight and obese.

## 1. Introduction

Type 2 diabetes mellitus (T2DM) is a complex metabolic disease characterized by continuous hyperglycemia in the absence of treatment. T2DM is the most common type of diabetes, affecting around 90% of diabetics, including beta-cell dysfunction and insulin resistance [1,2].

According to the International Diabetes Federation, the global prevalence of T2DM in 2019 was 9.3, 95% confidence interval (CI) (7.4–12.1), and it is projected to increase to 10.9%, 95% CI (8.4–14.1) by 2045, affecting 700.2 million people worldwide. This prevalence is projected to be higher in countries where economies move from low-to middle-income, revealing the social complexities that need to be taken into account when studying this disease [1]. T2DM represents a huge economic burden on health systems around the world, which can be measured by direct medical expenses, indirect expenses due to productivity losses, premature deaths, and deleterious effects of diabetes on the countries’ gross domestic product [1].

In 2015, the global spending on this health condition reached 1.33 trillion dollars, and by 2030 it is projected that total spending related to T2DM will reach 8.39 trillion dollars [3]. As a result of the urbanization process, 310.3 million people with T2DM live in urban areas, representing a prevalence of 10.8%, while in rural areas the prevalence reaches 7.2% [1]. This phenomenon is more evident in low- and middle-income countries [1,4].

Some risk factors have been associated with the onset of diabetes, such as overweight, obesity, and physical inactivity [5]. Other factors associated with T2DM prevalence are ethnicity, gender, socioeconomic status, age, malnutrition, glucose intolerance, and hypertension [5,6].

Structural and social determinants of health outcomes, such as income and social position, education level, working conditions, health services access, and physical environment, generate different conditions and exposures that have a clear impact on non-communicable chronic diseases such as a T2DM; and their recognition allows healthcare workers and stakeholders to perform in multidisciplinary spaces to design and implement public policies and health interventions in those conditions that can be modified [7,8,9,10], highlighting the importance of implementing prevention and health-promotion strategies at an early age.

Figure 1 describes the multilevel dependency of this phenomenon, considering the individual factors, neighborhood, municipal level, city, and country variables. This theoretical scheme has been proposed by the authors to explain the relationships of environmental, demographic, and health variables that can explain the unequal distribution of T2DM and give a theoretical base to the present systematic review.

Other risk factors have emerged to explain the socioeconomic gradients in the T2DM prevalence. Because of this, preventive measures emerge from the result of multisectoral work to reduce the prevalence of diabetes risk factor, in particular, the ones that can be modified—such as overweight, obesity, lack of physical activity, and an unhealthy diet—through a combination of fiscal policies, laws, regulation of marketing strategies, environmental policies, and urban changes; with emphasis on social and environmental determinants with a multisectoral, promotional, and preventive approach [11,12,13].

The major determinants of T2DM are socioeconomic factors that influence urban exposure. In this regard, the link between T2DM prevention and urbanization processes needs to be understood as an environmental health factor [4,8,13,14].According to a meta-analysis that studied the influence of the built environment in urban areas on T2DM, living in an urban area was associated with higher T2DM risk: odds ratio (OR) = 1.40, 95% CI (1.2–1.6), compared to living in rural areas. Moreover, the study highlighted that environmental characteristics such as neighborhood green spaces were associated with lower rates of T2DM [13]. In addition, exposure to green spaces have been associated with higher levels of physical activity, better physical and mental health, and lower stress, making it possible to achieve higher social capital [15,16]. In particular, green spaces have been studied and proposed as health determinants, because their distribution differs across populations and impacts the health status and wellbeing of the population [17,18,19].

In the urban context, green spaces can be defined as open spaces with natural elements such as parks, playgrounds, and recreation areas, both public and private, which can be used by the population for individual and social activities. These areas are known as urban land covered by vegetation [17,18]. It is well known that the urbanization process reduces time spent in contact with nature, but in the urban context, green spaces, forests, fields, street trees, and urban parks can play a protective role in the development of non-communicable diseases [19]. In this sense, the possible theoretical pathways of green space exposure and improvements on health are based on mitigation of air and noise pollution, because greenspaces are not sites of pollutant emission and also provide an acoustic barrier while reducing heat island effects, i.e., urban areas with more buildings than natural landscapes and have higher temperatures than outlying areas. Green spaces also reduce stress, increases positive emotions, and allow recovery from fatigue, thereby increase physical activity levels, and also provide a place for social contact [17,18,19,20].

Although this review is not a study about pandemics, we cannot disentangle ourselves from the potential impacts of green space exposure and urban policy modifications that must be discussed in light of the COVID-19 pandemic. In the COVID-19 pandemic context, diabetes and other non-communicable diseases have been confirmed to make people more vulnerable to getting a confirmed SARS-CoV-2 test and to dying because of complications of this virus [21,22]. Particularly, T2DM was associated with a higher incidence of COVID-19 relative risk (RR) = 2.38, CI (1.88–3.03) and mortality RR = 2.12, CI (1.44–3.11) [22]. Furthermore, T2DM and COVID-19 are called “socially transmitted diseases,” and it is well accepted that these conditions are driven by environmental factors, such as urban green spaces [21]. In addition, it is important to promote the maintenance of green spaces and accessible urban parks during COVID-19 (and any other pandemic) because it is beneficial for both physical and mental health, taking into account health recommendations to prevent transmission of COVID-19 [23].

In this regard, it is possible to infer that some of the modifiable structural conditions relevant in this context are urban planning policies that can be directed to allow everyone to live in healthier environments and have access to green spaces close to home. Unfortunately, access to green spaces and urban parks is not evenly distributed and reflects deep social inequalities in urban areas [24,25,26,27]. This evidence suggests that environmental factors such as green spaces can influence the appearance of T2DM; however, the limitations with regard to the quality and quantity of the studies do not allow us to infer causality [10].

The following study aims to know the magnitude of the effect that the exposure variable (green space) has on the prevalence of T2DM through the analysis of recent scientific evidence. It is relevant to mention that this research contributes to understanding the onset of T2DM from a socio-spatial point of view, with a focus on social determinants, and considering the biological mechanisms of the disease, such as the immunological pathways related to exposure to microbial inputs that show a particular inflammatory reaction due to lower levels of C-reactive protein and cytokines and high levels and greater resistance to stress [28].

To our knowledge, this is the first systematic review that studies the association between green space exposure in urban settings and T2DM (and some of its risk factors), also adding the value that it broadens the inclusion criteria to non-English languages such as Spanish and Portuguese, often neglected in systematic reviews.

The aim of this review is to study the relationship between green spaces in the urban context and their relationship with T2DM, obesity, and physical activity in the general population.

## 2. Materials and Methods

This systematic review was methodologically based on the Cochrane Handbook of Systematic Reviews [29] and the Preferred Reporting Items for Systematic Reviews and Meta-analysis (PRISMA) guidelines to report the structure [30,31]. The Appendix A shows the PRISMA checklist for reporting systematic reviews.The reviewers were trained by the lead author (F.D.l.F.), who received a formal course in this methodology. In particular, the reviewers were trained in extraction and data management and the critical appraisal of studies.

### 2.1. Data Sources and Search Strategy

Firstly, MeSH terms were identified in PubMed (United States National Library of Medicine, Montgomery, MD, USA) and used to build a search strategy. The MeSH term used as the main outcomes were “Type 2 diabetes”, “Diabetes mellitus, non-insulin-dependent”, and “prediabetes”. With the purpose of obtaining a better understanding of the phenomena, secondary outcomes were included, taking into account the conceptual framework of environmental factors associated with T2DM based on IMAGINE 1 [10]. MeSH terms used as secondary outcomes were “Obesity”, “Overweight”, and “Physical Activity”. Terms used for determining exposure were “Green space”, “Green spaces”, and “Built environment”.

Three databases were used to identify potential articles. The PubMed, EMBASE, and LILACS databases were used, LILACS contains literature in Spanish and Portuguese and is an important repository in Latin America. The search strategy was performed using the following MeSH terms in each database:

PubMed; ((((type 2 diabetes mellitus [MeSH terms]) OR diabetes mellitus, non-insulin-dependent [MeSH terms]) OR diabetes mellitus [MeSH terms]) OR prediabetes [MeSH terms]) AND green space,

((((type 2 diabetes mellitus [MeSH terms]) OR diabetes mellitus, non-insulin-dependent [MeSH terms]) OR diabetes mellitus [MeSH terms]) OR prediabetes [MeSH terms]) AND green spaces,

(((((type 2 diabetes mellitus [MeSH terms]) OR diabetes mellitus [MeSH terms]) OR non-insulin-dependent diabetes mellitus [MeSH terms]) OR prediabetes [MeSH terms])) AND built environment,

((obesity [MeSH terms]) OR overweight [MeSH terms]) AND green space,

((obesity [MeSH terms]) OR overweight [MeSH terms]) AND green spaces).

EMBASE (((type 2 diabetes mellitus [MeSH terms]) OR diabetes mellitus, non-insulin-dependent [MeSH terms]) OR diabetes mellitus [MeSH terms]) OR prediabetes [MeSH terms]) AND green space,

(“non-insulin-dependent diabetes mellitus” OR “impaired glucose tolerance”) AND “built environment”,

(obesity AND “green space”).

LILACS: “DIABETES” [Palabras] and “AREAS VERDES” [Palabras],

“non-insulin-dependent diabetes mellitus” OR “impaired glucose tolerance”) AND “built environment”,

obesity AND “green space.”

### 2.2. Selection Criteria

The inclusion criteria on this systematic review were as follows: articles written in Spanish, English, or Portuguese. Published since the year 2009 to October 2020. The type of studies included were observational studies such as case–control, cohort, and cross-sectional studies. The only exposition included was green spaces. Studies with T2DM as the primary outcome were included. As a secondary result, articles with obesity, overweight, and physical activity as health outcomes were also included. The results of each of the studies are summarized in Table 1.

### 2.3. Data Extraction and Management

Figure 2 was based on the PRISMA statement illustrating the process for identifying and selecting items. Firstly, by using a spreadsheets program (Google Sheets), potential articles to include were listed. Secondly, the title and abstract were independently selected by pairs of reviewers (MB–NC, MS–GM, FD–CC). Disagreements were resolved by a third reviewer. The selected articles were included, and the following information was extracted: (1) author, (2) country, (3) study design, (4) the number of participants, (5) sex, (6) age of participants, (7) exposure, (8) exposure assessment, (9) outcome, (10) outcome assessment, (11) effect size, and (12) quality assessment. Table 1 shows the characteristics of the included studies.

### 2.4. Quality Assessment in Included Studies

For the analysis of quality and risk of bias, The National Institutes of Health’s Quality Assessment Tool for Observational Cohort and Cross-sectional Studies was used according to each type of article, as it has been used in others systematic review [32]. This tool has a total of 14 items to evaluate; each completed item gives 1 point. As a result, three categories were created by the authors to score the quality assessment: good (13–14 points), fair (11–12 points), and bad (<10 points). The studies’ scores can be found in Table 1 and Table 2.

## 3. Results

Figure 2 shows the PRISMA flow diagram for the search, identification, and selection of the articles included in this systematic review. Firstly, 161 articles were found on PubMed and Embase; no article was found on the LILACS database. Secondly, 10 articles were included from other sources. From a total of 171 articles, 26 were duplicate and were removed. Of the remaining 145 articles, title and abstract were reviewed. As a result, 32 articles were excluded. A total of 113 articles were eligible for full-text review. After reviewing the full text, 94 articles were excluded. A total of 19 scientific articles were included in this systematic review.

As a result of this systematic review, 19 scientific articles were included. The outcomes related to greenspace exposure will be represented by categories. Firstly, T2DM such as main outcome and then intermediate health outcomes (overweight and obesity, physical activity).

### 3.1. Diabetes

Seven scientific articles that directly evaluate green space exposure and T2DM as an outcome were found. All articles studied participants 15 to 85 years old [33,34,35,36,37,38,39]. The exposure to open and public green spaces was measured with the normalized difference vegetation index (NDVI) and geographic information system [35,36]. Moreover, subjective indicators were used, using surveys (Bhzad et al.). The diabetes outcome was measured using health databases (Roland et al.) and medical records. Two cohort studies showed a protective role of green spaces related to T2DM, resulting in an adjusted OR of 0.9, CI (0.87–0.93) and a relative risk of 0.75, CI (0.69–0.83) [35,36]. Another cohort study conducted on 23,865 individuals showed that people living closer to green spaces had a lower risk to develop T2DM when comparing to people living far away [38]. People who lived further away from green spaces with sports amenities had more risk of T2DM, with a prevalence risk ratio of 1.09, 95% CI (1.03–1.11), *p* < 0.01, compared with those who lived closer [33,34] In the spatial analysis, areas with higher distance to green spaces with sports facilities evidenced higher prevalence of T2DM. This relationship is plausible because proximity to green spaces can stimulate physical activity and then protect against T2DM [33]. Finally, after the adjustments for sex, age, and socioeconomic status, among other factors, participants living in a neighborhood with more availability of integrated green spaces had a lower risk of T2DM [37,38,39].

### 3.2. Overweight and Obesity

In the selected studies, approximately 3 million people participated in cohort and cross-sectional studies conducted in countries with high economic development levels. In these studies, obesity and overweight were evaluated as measured by body mass index (BMI) and self-reports [37,40,42,44,51].The researchers from primary studies were interested in highlighting the exposure to environmental factors at different age groups, evidencing the importance of green space exposure since an early stage of development and throughout the entire life cycle. The studies that included children between 3 and 13 years of age highlight that access to green areas in the neighborhood were associated with a lower prevalence of obesity [43,49,50]. The results obtained from medical records, self-declared health surveys, and standardized questionnaires allowed the construction of mathematical models adjusted for sociodemographic variables, showing a significant association between green spaces in the neighborhood and obesity or overweight. For example, children who lived in areas with less exposure to green areas had a higher risk of obesity, OR = 1.72, CI (1.15–2.26), *p* < 0.05, compared to those who lived in a neighborhood with more exposure to green spaces [49].

Among the studies carried out in a large group of adults over 18 years old, it stands out that compared to exposure to green areas and other characteristics of the neighborhood, living closer to open green spaces was associated with a lower degree of overweight and obesity [37,42,46,51]. These results were obtained from self-reported surveys, administrative records, georeferenced data, and multivariate analysis models adjusted for sociodemographic variables. The impacts of the findings could be different depending on the gender of the participants and the geography of the residential locations. However, other studies failed to find significant associations between green spaces and obesity or overweight [45,47,48].

### 3.3. Physical Activity

Around 400,000 people aged 2 to 70 years old, living in countries with high levels of economic development, participated in four cross-sectional studies and one longitudinal study. The four cross-sectional studies included participants 18 years and older, and the longitudinal one followed 2–5-year-old preschool children from low-income families in New York City (Lovasi et al.). These five studies scored a fair-to-good result in the quality assessment (11–13 from 14).

In these investigations, exposure to green areas and access to open spaces for walking or physical activity were evaluated (41–43,48,51). These studies used different data collection strategies such as population census or georeferenced data through self-reported questionnaires and sensitive measurement instruments. The analyses were made adjusting for sociodemographic variables, allowing main results to conclude that the vast majority of people studied who had access to green areas in their neighborhood effectively use these areas and increase the likelihood of practicing physical activity [33,34,45,48]. One of these studies (Prince et al.), however, reported gender differences, with men reporting lower physical activity associated with higher green space areas, and women showing no significant associations between social-environmental variables and physical activity, confirming the complex nature of the phenomenon and the need to include a gender approach in the analysis. Ultimately, being exposed to green spaces has been proved as a promoter of physical activity in adults and preschoolers, in terms of duration and frequency of activities, with a positive association between these variables [41].

## 4. Discussion

This systematic review aimed to analyze the scientific evidence regarding green space exposures and T2DM as the main health outcome and obesity and physical activity as secondary outcomes. Different definitions of green spaces and greenness measurements were found in this study. Green space exposure had different measurement methods, such as distance to parks, normalized difference vegetation index (NDVI)—used to measure living vegetation by the reflectance levels that the vegetation emits from the photosynthesis process—the density of trees in the neighborhood, park areas per km^2^, geographic information system, postal code use, and by self-reported questionnaires. There is no consensus yet about which kind of methodology is best for answering this research question [38]. It is important to discuss the methodology spectrum to assess the validity measure of green spaces. Thus, subjective measures such as questionaries and self-reported are usually of lower cost and easier to implement than objective measurements such as direct assessment of green spaces, e.g., NVDI and other georeference methods.

T2DM was measured by medical records, surveys, and blood tests. Secondary outcomes such as physical activity and obesity were included in this systematic review due to the causal pathway between these risk factors regarding T2DM.

The relationship between greenness or greenspaces has been studied in differing contexts. It suggests that green spaces play a key protective role against air pollution, allowing to avoid chronic inflammation processes [52]. The causal pathway to understanding this relationship also includes demographic factors such as age, gender, ethnicity, and socioeconomic status and living context, such as cultural factors, safety and infrastructure, local and regional policy, and rural or urban setting. These factors can moderate the opportunities or barriers to the use of green spaces, which if they are available, promote relaxation activities, encourage physical activity, interaction with nature, and social interactions within greenspaces and participation in group activities [53]. Furthermore, neighborhood characteristics, including green areas, can promote or strengthen the city’s social capital [54]. This systematic review results in agreement with other studies that focus on the built environment, including green spaces and health-related outcomes, highlighting the urban context’s role in chronic conditions and signaling the importance of addressing inequalities to allow a more homogeneous distribution of the urban green spaces [10]. Other studies have proposed the effect of the green spaces as promoters of endocrinological effects and highlighted the role of nature in the inflammatory response and chronic conditions [28]. There is also evidence in the same causal direction of this review, supporting the idea that vegetation in the urban context contributes to improving human health and social well-being, by showing that the majority of people exposed to green spaces had a smaller risk of having T2DM and other risk factors such as obesity and sedentarism [55].

Regarding the available evidence, more primary studies need to be conducted, considering the type of green space exposure and the measurement of main outcomes such as diabetes but also confounders, risk factors, precursors, and effect modifiers, to properly isolate the effect. It would be relevant that these studies include a gender approach since some results of this systematic review showed that there are significant gender differences to be considered. In the same direction, ethnicity and other determinants should be considered when analyzing the results.

Further research needs to be carried out to determine the social and biological links between green spaces and type 2 diabetes mellitus, especially in low- and middle-income countries, to identify potential barriers of use, inequities in distribution, and to encourage stakeholders to generate public policies where urban and health factors are considered to reduce health impacts of chronic conditions such as T2DM.

Of the 19 studies included in this systematic review, using a characterization defined by the authors based on The National Institutes of Health´s Quality Assessment Tool for Observational Cohort and Cross-sectional Studies, seven articles were found to be good, seven fair, and five were found to be poor in the quality assessment. Because of this, results and recommendations need to be addressed with precaution.

One of the limitations that this study was hoping to address was publication language bias because of the lack of non-English primary articles incorporated in other systematic reviews. Sadly, the search strategy used in the present study did not return any non-English articles. On the same subject, the search strategy failed to find articles conducted in Latin America and Africa regions. Both of these issues could demonstrate the importance of conducting primary studies in those regions, also taking into consideration that both of these regions present high levels of inequality.

The current urbanization process must consider the distribution and protection of urban green spaces, given a growing body of literature that shows its impacts on non-communicable diseases (NCDs). When analyzing the role of green spaces in current urban processes, it is necessary to recognize their impact on public health, given the fact that they impose a positive effect on the health of nearby residents not only in terms of scale, function, and accessibility but also in vegetation cover, ecological dimension, and social and environmental quality.

Different types of green spaces should be a matter of interest for urban policymakers, in terms of prioritizing and safeguarding these areas, at a time when more research shows the benefits of green spaces, especially when there are economic actors with greater interest in large-scale real estate projects without considering protecting the exposures to green spaces.

## 5. Conclusions

There is significant evidence supporting the protective role of green spaces in the urban context against T2DM and other chronic health conditions such as obesity and sedentarism. People and communities living in neighborhoods with more green spaces and closer to parks with sports facilities had less risk of having T2DM. The exposure to green spaces also reduced the likelihood to be obese and boosted the probability to perform physical activity.

In times of the COVID-19 pandemic, communities’ access is more restricted to proximal green spaces, thus resulting in poor air quality and high rates of respiratory diseases and other health outcomes, making neighborhoods more vulnerable to poorer health outcomes and to being disproportionately harmed by health costs and economic and social aspects of the COVID-19 pandemic that underlie the health conditions of those neighborhoods [56].

## Figures and Tables

**Figure 1 ijerph-18-00097-f001:**
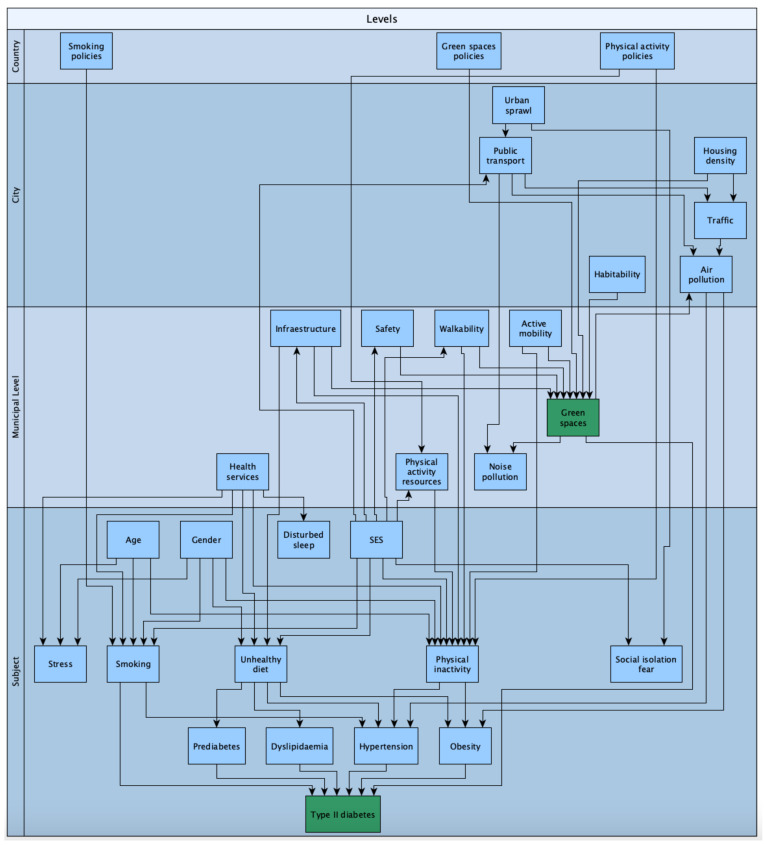
Scheme of multilevel dependency on green-space exposure and related variables. Source: own elaboration.

**Figure 2 ijerph-18-00097-f002:**
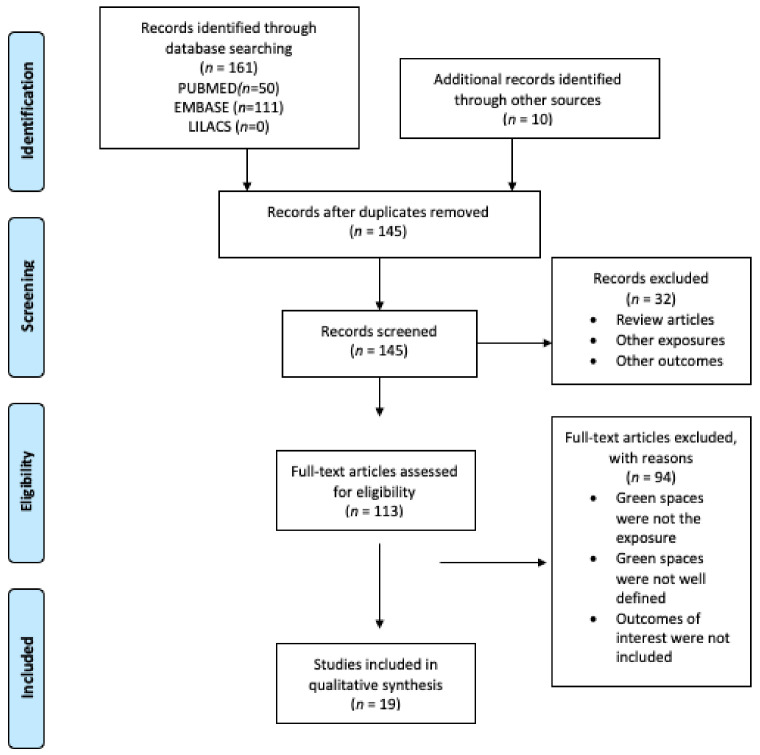
Preferred Reporting Items for Systematic Reviews and Meta-analysis (PRISMA) flow diagram of the screening process.

**Table 1 ijerph-18-00097-t001:** Characteristics of included studies. Principal outcome: type 2 diabetes mellitus (T2DM).

No.	Author, Year, Location	Study Design	Population Description	Exposure	Exposure Assessment	Outcome	Outcome Assessment	Effect Size/Association	Factors Adjusted	Quality Assessment
1	Ngom et al., 2016, Canada [33]	Cross sectional study	*n* = 3,920,000, male and females >20 years old	Green spaces	Geographic information system, postal code	Diabetes	Health databases from surveillance	People who live further from green spaces with sports facilities have a prevalence rate ratio = 1.09 (1.03–1.13), *p* < 0.01	Age and gender	10/14
2	Sidawi et al., 2014, Kingdom of Saudi Arabia [34]	Cross sectional study	*n* = 76 male and female, 15–70 years old	Build environment- recreation and sport centers	Questionnaire about home environmental conditions	Diabetes	Medical records	Respondents, who had diabetes earlier, said that the recreation and sport centers are farther from their homes than those who had diabetes later	Socioeconomic conditions of the neighborhood	5/14
3	Clark et al., 2017, Canada [35]	Cohort	*n* = 380,378 males and females 45–85 years old	Green spaces	Normalized difference vegetation index (NVDI)	Diabetes	Medical records	Exposure to green spaces is protective against T2DM; adjusted odds ratio = 0.9, CI (0.87–0.93)	Age, gender, area-level household income	11/14
4	Paquet et al., 2014, Australia [36]	Cohort	*n* = 3145 >18 years old	Public open spaces with green spaces	Normalized difference vegetation index (NVDI)	Diabetes, prediabetes	Medical records by fasting blood samples	People who have more access to open public spaces with green spaces have a relative risk = 0.75 (0.69–0.83), *p* < 0.001	Age, gender, education, household income, and area-level deprivation	13/14
5	Lee et al., 2015, Korea [37]	Cross sectional study	*n* = 16,178 47.50 mean age	Area of parks in neighborhood (km^2^)	Geographic information system	Diabetes, obesity, abdominal obesity	Medical records	People who live in a community with more parks areas in neighborhood have a lower risk of diabetes, OR = 0.86 (0.75–0.99); obesity, OR = 0.97(0.90–1.04); and abdominal obesity, OR = 0.83 (0.77–0.91)	Age, sex, smoking status, drinking status, and income level	12/14
6	Dalton et al., 2016, United Kingdom [38]	Cohort	*n* = 23,865mean age 59.1 years old	Neighborhood green space	Zip code- Geographic information system	Diabetes	Survey, medical records, hospital data	Individuals living in the greenest district quartile had a lower risk of developing diabetes, hazard ratio = 0.81 (0.65–0.9), *p* = 0.042	Sex, age, body mass index (BMI), parental diabetes, and socioeconomic status	13/14
7	Bodicoat et al., 2014, United Kingdom [39]	Cross sectional study	*n* = 1047 20–75 years old	Neighborhood green space	Zip code- geographic information system	Diabetes	Medical records by oral glucose tolerance test	For diabetes prevalence, the OR = 0.97 (0.80–1.17), 0.78 (0.62–0.98), and 0.67 (0.49–0.93) for increasing quartiles of neighborhood greenspace compared with the lowest quartile	Age, sex, ethnicity, area social deprivation score, urban/rural status	13/14

**Table 2 ijerph-18-00097-t002:** Characteristics of included studies. Secondary outcomes: obesity, overweight, and physical activity.

No.	Author, Year, Location	Study Design	Population Description	Exposure	Exposure Assessment	Outcome	Outcome Assessment	Effect Size/Association	Factors Adjusted	Quality Assessment
1	Lovasi et al., 2013, USA [40]	Cross-sectional study	*n* = 11,562 children, 3–5 years old	Green spaces in neighborhood	Density of trees and park area per km^2^ using ZIP code	Obesity	Body mass index (BMI) z-score by health care provider	Density of street trees, β = −0.02 CI (−0.08, 0.03); prevalence ratio (PR) = 0.88 (0.79, 0.99) Area covered by parks, β = −0.01 (−0.03–0.01); PR = 0.99 (0.94–1.04)	Sex, race/ethnicity, age, and neighborhood characteristics	10/14
2	Shanahan et al., 2016, Australia [41]	Cross-sectional study	*n* = 1538, 18–70 years old	Frequency and intensity of exposure to nature	Self-reported by questionnaire/survey using the Nature Relatedness Scale	Physical activity	Number of days exercised for 30 minutes or more per week	Nature experience duration β = 0.19, *p* < 0.001; nature experience frequency β = 0.16, *p* < 0.001	Age, gender, income, children in home, neighborhood disadvantage, workday/week, highest qualification, ethnicity, BMI, social cohesion	11/14
3	Prince et al., 2011, Canada [42]	Cross-sectional study	*n* = 3883, males and female, >18 years old	Green spaces and park areas	Geographic information system, geocode	Obesity Physical activity (PA)	Obesity = BMI measurement PA = Self-reported by questionnaire	Physical activity was lower for men in neighborhoods with a higher green space area, odds ratio (OR) = 0.93, 95% CI (0.87, 0.9). For females, green spaces were protective of being obese or overweight, OR = 0.67 CI (0.54–0.84)	Sex, age, socioeconomic status, social and built environment characteristics	12/14
4	Lovasi et al., 2011, USA [43]	Longitudinal study	*n* = 428, 2–5 years old, males and females	Green spaces	Street tree density by geographic database	Physical activity	Accelerometer	Land use mix was associated with physical activity (26 more activity counts/minute per standard deviation increase in mixed land use, *p* = 0.015)	Age, sex, and race/ethnicity, mother (age, born outside of the USA, use of Spanish, employed/student status), household (number of rooms), the total number of hours recorded as awake, the time of year	13/14
5	Hrudey et al., 2015, Netherlands [44]	Cohort	*n* = 3469, 5–6 years old	Green spaces	Survey with Likert scale of green spaces satisfaction	Obesity and overweight	Self-reported	No significant association was found, after adjusting for variables. β = −0.002, CI 95% (−0.3–0.3)	Maternal pre-pregnancy BMI, maternal smoking during pregnancy (yes, no), duration of exclusive breastfeeding (<3 months, 3–6 months, ≥6 months), and age at introduction of solid foods (<4 months, ≥4 months), Maternal education and maternal BMI	13/14
6	Sanders et al., 2015, Australia [45]	Cohort	*n* = 4423, 6–13 years old	Green spaces	Proportion of green spaces available in neighborhood by postcode	Obesity	Face-to-face interview, waist circumference (WC), and waist-to-height ratio (WtHR)	Compared to those who have 0% to 5% green spaces at the local level, children with >40% green space tended to have lower WC (β boys, −1.15, 95% CI −2.44, 0.14; β girls, −0.21, 95% CI −1.47, 1.05) and WtHR (β boys, −0.82, 95% CI −1.65, 0.01; β girls, −0.32, 95% CI −1.13, 0.49). No statistically significant results were found	Sex, age, socio economic status	10/14
10	Putrit et al., 2015, USA [46]	Cross-sectional study	*n* = 9971, >18 years old	Green spaces, parking facilities	Self-reported survey	Obesity/overweight	Self-reported	People who perceived more availability of green spaces showed odds ratio = 0.84, CI (0.72–0.97) for obesity and OR = 1.08, CI (0.98–1.20) for overweight. After adjusting for age, the effect size, for people from 40 to ≤65 OR for obesity = 0.80, CI (0.66–0.96), and >65 years old OR = 0.71, CI (0.54–0.93)	Age, gender, educational level	13/14
11	James et al., 2017, USA [47]	Cohort	*n* = 23,435 women, 60–87 years old	Green spaces	Normalized difference vegetation Index	Obesity	Self-reported weight and height	No significant association between all variables in the model and BMI 0.01% (−0.36–0.37)	Age, race, smoking status, husband’s education level	10/14
12	Klompmaker et al., 2018, Netherlands [48]	Cross-sectional study	*n* = 387,195, >19 years old	Green spaces	Distance to the nearest park and normalized difference vegetation index	Obesity Physical activity	Self- reported	No significant association was found, within 100 m of a park compared to the reference category (>1000 m) where 1.04 (95% CI: 0.83–1.25) and 1.02 (95% CI: 0.96– 1.07) for the highly urban and moderate–low urban population, respectively. For the elderly (≥65 years) and non-elderly, these odds ratios were 1.01 (95% CI 0.96–1.07) and 1.02 (95% CI: 0.94–1.08), respectively. Physical activity was higher in people who lived closer to the park entrance, odds ratio = 1.08 (1.03–1.14). For NVDI, greenness increased the OR = 1.14 (1.10–1.17) in the highest quintile compared to that in the lowest.	Age, sex, socioeconomic status, marital status, country of origin, work, household income, level of education, smoking status, alcohol use, indoor physical activity	12/14
13	Petraviciene et al., 2018, Lithuania [49]	Cross-sectional study	*n* = 1489 mothers and their 4–6-year-old children	Green spaces	Normalized difference vegetation Index	Obesity and overweight	Self-reported by standardized questionnaires	Children who live in areas with less greenness exposure, have higher risk of being obese/overweight OR = 1.72 CI (1.15–2.60), *p* < 0.05	Family status, maternal age, education, employment status, smoking during pregnancy, secondhand smoking, mother–child relationship, NO_2_; child´s sex, birth weight, and sedentary behavior	12/14
14	Dadvand et al., 2014, Spain [50]	Cross-sectional study	*n* = 3178, 9–12 years old	Green spaces	Normalized difference vegetation index, proximity to green space by Urban Atlas Map	Obesity	Self-reported by questionnaire	In relation to 4 buffers of green spaces: 100 m buffer and obesity odds ratio (OR) = 0.32, CI (0.75–0.93), 250 m buffer OR = 0.81, CI (0.71–0.92), 500 m buffer OR = 0.83, CI (0.78–0.98)	Parental education, type of school, sport activity, and having siblings	12/14
15	Coombes et al., 2010, England [51]	Cross-sectional study	*n* = 6803, >18 years old	Green spaces	Geographic information system, geocoding	Obesity Physical Activity	Self-reported by questionnaire	Respondents who visit green spaces with less frequency showed odds ratio = 0.39, CI (0.33–0.45), *p* < 0.01 of achieved physical activity guidelines and odds ratio = 1.44, CI (1.25–1.66) of being obese or overweight	Age, sex, socioeconomic status, self-rated health, area-level deprivation	13/14

## Data Availability

Data sharing not applicable, because this study is a systematic review.

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
