# Peer review of "Green Space Exposure Association with Type 2 Diabetes Mellitus, Physical Activity, and Obesity: A Systematic Review"

_ijerph, 2020, doi:10.3390/ijerph18010097_

Round 1
Reviewer 1 Report
The manuscript submitted by Dr. De la Fuente and colleagues deals with the issue of the association between urban exposure to green spaces and the risk of type 2 diabetes. In a systematic review the Authors try to put a very intuitive subject of beneficial influence of greenery for human health into some more scientifically-based framework. Their paper is of considerable interest for public health although some minor issues need to be addressed before the manuscript is acceptable for publication.
First of all, the title could be more informative - not just "a systematic review" but maybe containing some precise concluding message. Then, the Image 1 is hardly legible - it actually shows the complexity of various factors which often cross-react and affect type 2 diabetes prevalence, but it would look much better in a horizontal layout. Likewise, Table 1 needs to be revised in order to make it clearer - maybe a horizontal layout would help? maybe some abbreviations? In line 115 the abbreviation DMII is used instead of T2DM, which is used elsewhere across the manuscript - just to be consistent with the abbreviations. In the Discussion section the notion of "normalized difference vegetation index" could be briefly explained as healthcare readers who are less familiar with methods of green spaces quantification, can feel uncomfortable. Furthermore, in line 217 the statement "living closer to green spaces is associated with lower overweight and obesity" sounds odd, since overweight and obesity are quantitative features - maybe "lower degree of overweight and obesity" would be better? Finally, there are some very minor language mistakes - typos and small grammar issues that should be corrected.
Overall, this is an interesting paper, with some very current references to pandemic state as well. It deserves further consideration after some minor revision.
Reviewer 2 Report
This review summarized the current evidence of the association between green spaces exposure and diabetes prevalence in the general urban population. Although authors did a good job, some issues should be figured out before the paper is accepted. Please see below.
- Please add more keywords in order to increase the possibility that more reader can search your paper.
- Line 29, T2MD is a typo; Line 32, please specify the abbreviation CI.
- There are so many paragraphs in the section of Introduction. Please do an appropriate combination.
- In the last part of Introduction, please add the significance and novelty of the study.
- Line 49, Figure 1 should be better.
- Line 144, please show your Figures in order.
- Line 156, please give more explanation about Figure 1.
- Please give several subtitles for the part of Results, such as 3.1, 3.2, …
- Please give more discussion, and a condensed conclusion. Moreover, a supplement of future perspectives would be helpful for the manuscript.
- In my opinion, a review paper usually does not contain the section of methods and discussion. Please double-check and follow the style of the journal IJERPH.
Reviewer 3 Report
Green space exposure and Type 2 Diabetes Mellitus: a systematic review
The topic is relevant however I have many concerns about the methods, results and conclusion of this study.
Abstract - It is necessary to include methods information. On this version, the methods are not enough. Conclusion should only respond the aim and not include recommendations. Key words - add obesity and physical activity.
Introduction
Introduction is too long and each paragraph should be more well-structured and following the scientific writing recommendations.
Include specialized and updated literature.
The authors must improve the relationship between the two conditions also to justify the manuscript in a more consistent way.
Image 1: Scheme should be part of results of this manuscript and could be include as the aim. The way of the scheme is presented is complicated because the reads cannot understand the methodology and base on what previous knowledge and articles the image 1 had been elaborated. “This theoretical scheme has been proposed by the authors regarding some environmental, demographics and health variables that can explain the unequal distribution of T2DM” (lines 46-48). Then, it is not part of the presented manuscript and do not help to elucidate the justify of this literature review.
Are there previous systematic reviews on this theme? Which ones? Why these ones are not enough?
Lines 120-121 – “to analyze the published evidence regarding green spaces in the urban context”. It is methodology and not the aim (objective). The last paragraph of introduction must let clear the aim of the study.
Methodology
It is important to clarify the definition of green spaces and how it could be assessment in a validity way.
It is also essential to define if there is a specific range of years old? It can be complicated to include on the same analyses children, adolescent, adults and older adults.
Walkability is also an exposure to be evaluate?
Is it study with focus on T2DM?
How obesity can be assessment? Visceral obesity also can be included?
Have the reviews been trained? How?
Which software have been used to manager the list of potential articles?
Include information of study quality assessment and risk of bias
Please, described the selection strategy and use of terms to find the articles on the properly way of systematic review.
Why GRADE and Dows and Black tools were not applied?
Line 139 – How case-control and clinical trials studies can be a source of information for this study? Which kind of information had extract of this articles?
Lack of information regarding data analysis.
Results– on the abstract the aim is “the association between green spaces exposure, diabetes prevalence and other intermediate outcomes in the general urban population” however it is not possible to visualize it on the results.
Figure 1 – the information of full-text excluded articles are on the wrong position
Several outcomes on the same tables let the information obscure. I recommend a table with T2DM and other with obesity.
Table 1 should be improved. It seems more an extraction table than an edited table to include in an article. It is hard to understand this table. Consequently, the affirmations on line 185 till 233 is not possible to see on the table.
Physical activities are also an outcome? It was not defined on methods. Same question for blood pressure.
On this study seems that the focus is T2 DM however on table 1 there are six studies with children. The authors should amend it.
Discussion
It is poor and to not follow the principles of scientific writing and PRISMA-P.
Conclusion
How the results are not clear and mix several information it seems that the conclusion is not supported by the results.
It is not adequate do mix conclusion (to respond the objective) with recommendations.
Reviewer 4 Report
Peer Review: “Green space exposure and Type 2 Diabetes Mellitus: a systemic review”
Journal: International Journal of Environmental Research and Public Health
Recommendation: Accept with minor revision
In this paper, the De la Fuente et al performed a systematic review of publications studying correlation between access to urban green spaces, and the occurrence of type 2 diabetes mellitus in a population. The objective is to review the current evidences of the association between urban green spaces exposure and prevalence of Type 2 Diabetes in a population level. Systematic search was performed on three databases including PubMed, Embase, and Lilacs, resulting to 19 papers being included. The De la Fuente et al then summarized the results from these 19 papers, focusing on the prevalence on Type 2 diabetes, obesity and physical activity.
Minor points
- In mentioning the studies/papers being referenced, it would help to name which studies or papers these are. For example, Line 185 “Seven scientific articles that evaluate directly green spaces exposure and T2DM as an outcome were found. (39-41,52-55)”. It would help to mention the first authors of these seven studies in referencing them, and giving a summary and more details as to what these studies have in common and what these studies have different measured
- There are inherent limitations to population-based studies, such as ethnicity of the population. Certain ethnicities are known to have high predisposition towards development of Type 2 Diabetes and obesity. However, only four or five of the papers being reviewed had this factor adjusted. Mentions of the limitations of some of these studies, and caution into their interpretation should be included in the review.
Round 2
Reviewer 2 Report
Thank you for your revised manuscript. The quality of the manuscript is significantly improved. I have no further comments.